# ACCELERATING DISCRETE DIFFUSION DECODING WITH PARALLEL SCAN

## ABSTRACT

Diffusion-based language models provide strong controllability and parallel generation capabilities, but suffer from prohibitively high decoding cost. Block diffusion, a semi-autoregressive approach, alleviates this issue by reducing diffusion steps and enabling KVCache utilization, yet it restricts parallel decoding strictly within blocks, preventing inter-block parallelism. In this work, we identify a class of *associative states* in diffusion models: **blocks that can be independently sampled without conditioning on the prefix, and later refined once the prefix becomes available, effectively performing a form of self-refinement**. Leveraging this property and inspired by classic parallel algorithms, we propose a novel **parallel-scan based decoding** framework. Our method incorporates two key techniques—*local remasking* and *global aggregation* —to enhance stability and efficiency. We further introduce a systematic design space of prefix-network topologies, cache strategies, and parameter configurations, and conduct a preliminary exploration of this search space. Empirically, our training-free approach achieves up to **68 tokens/s** throughput and **60.7% accuracy on GSM8K**, and we validate the effectiveness of the proposed techniques through ablations. Compared to mainstream semi-autoregressive methods, our results demonstrate that parallel decoding with structured parallel patterns remains a promising and underexplored direction for efficient inference in diffusion-based LLMs.

## 1 INTRODUCTION

Diffusion models have recently extended their success from continuous domains such as images and videos to discrete data (Austin et al., 2023; Campbell et al., 2022; Lou et al., 2023; Gat et al., 2024; Sahoo et al., 2024), driving rapid progress in text generation and language modeling. Only very recently have large-scale efforts demonstrated that diffusion-based language models (Nie et al., 2025; Ye et al., 2025) can approach, and in some cases match, the performance of strong AR baselines. For example, systems such as Seed Diffusion (Song et al., 2025b), Gemini Diffusion (Google Deep-Mind, 2025), and Mercury (Labs et al., 2025) provide compelling evidence that discrete diffusion can be scaled effectively, achieving competitive results on language modeling benchmarks. Despite recent progress, open-source diffusion language models remain inefficient: their denoising steps are inherently sequential, and each step requires high-cost bidirectional attention. Consequently, decoding is much slower than in autoregressive models, making inference efficiency the key barrier to their practical adoption at scale.

A major line of work has sought to mitigate the inefficiency of open-source diffusion language models. Existing approaches can be roughly categorized into two families. The first modifies the decoding paradigm itself. Block diffusion (Arriola et al., 2025) introduces a semi-autoregressive structure that enables KVCache across blocks. ESO-LLM (Sahoo et al., 2025) extends this idea by mixing intra-block and inter-block decoding strategies. The second line exploits KVCache reuse: since the key and value representations between adjacent denoising steps are often nearly identical, they can be cached and shared across steps, substantially reducing redundant computation (Wu et al., 2025; Ma et al., 2025; Song et al., 2025a). These methods all share a common goal: to reduce computational overhead and lower inference latency.

However, the reliance on KVCache inevitably constrains one of diffusion's most distinctive advantages: the ability to decode in a fully parallel manner. By enforcing sequential dependencies across

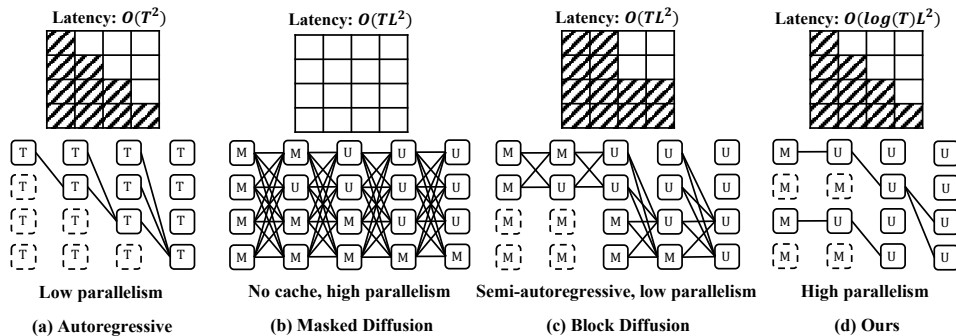

Figure 1: **Decoding Patterns.** Given a current sequence of length $L$, an LLM aims to predict the $T$th token, while a diffusion model predicts the $T$th denoised token. (a) Autoregressive models exhibit quadratic complexity $O(T^2)$ with low parallelism. (b) Standard diffusion models suffer from cubic complexity $O(TL^2)$ since they are incompatible with KVCache and must recompute attention states at every step, despite allowing full-sequence parallelism. (c) Block diffusion reduces the cost by enabling KVCache reuse, but the semi-autoregressive structure introduces sequential dependencies, limiting parallelism across blocks. (d) Our method promotes high parallelism by restructuring diffusion decoding into a prefix-scan style process, while maintaining KVCache compatibility. This reduces the effective latency to $O(\log(T) L^2)$, enabling efficient large-scale inference with both stable convergence and improved throughput. T: Tokens, M: Masked tokens, U: Unmasked tokens.

or within blocks, existing methods sacrifice the intrinsic parallelism of diffusion in exchange for cache efficiency. This trade-off raises a natural question: rather than only optimizing for latency, can we instead uncover and exploit the inherent parallelism inside diffusion models to fundamentally improve decoding throughput? We ask whether the decoding process itself can be restructured to expose richer forms of parallelism.

Our key observation is the existence of *associative states*—positions that can be decoded speculatively and later revised once additional context becomes available, exhibiting a form of self-refinement. This property opens the door to structured parallel decoding inspired by classical algorithms such as prefix-scan. Crucially, rather than applying prefix-scan naively, we introduce two mechanisms tailored to diffusion: *local remasking*, allowing the model to revise early drafts through selective uncertainty, and *global aggregation*, enabling remasked tokens to interact with the full KV-Cache for stable convergence. Together, these techniques transform diffusion decoding from a rigid sequential pipeline into a flexible parallel process, shifting the design goal from latency reduction to throughput maximization. In summary, our key contributions include:

- We identify **associative states**, which enable speculative decoding followed by refinement, offering a new source of parallelism beyond semi-autoregressive approaches.

- We propose a **parallel-scan based decoding framework** that restructures diffusion decoding into parallelizable stages, augmented with two mechanisms—**local remasking** for token refinement and **global aggregation** to ensure stable convergence.

- We establish a systematic design space encompassing prefix-network topologies, cache strategies, and key parameters, and conduct an empirical study showing that our training-free method achieves up to 68 tokens/s throughput with 60.7% accuracy on GSM8K.

## 2 RELATED WORK

### 2.1 DISCRETE DIFFUSION LANGUAGE MODEL

Diffusion models on discrete data were first formalized by D3PM (Austin et al., 2023), and later extended to more general formulations such as CTCM (Campbell et al., 2022), score-based methods (Lou et al., 2023), and flow matching (Gat et al., 2024). Subsequent studies observed that, compared to uniform corruption, masked transition matrices provide stronger performance. Recent work further simplified the training process and LLaDA (Nie et al., 2025), Dream (Ye et al., 2025), etc demonstrated effectiveness at larger model scales.

Consider a sequence of length $L$: $x = (x_1, x_2, \ldots, x_L)$, where each token $x_i \in \mathcal{V}$. The extended vocabulary $\mathcal{V}$ includes a special mask state $m$. The general forward noising process is a Markov chain: $q(x_t \mid x_{t-1}) = \text{Cat}(x_t; Q_t x_{t-1})$, where $Q_t \in \mathbb{R}^{|\mathcal{V}| \times |\mathcal{V}|}$ is the transition matrix at step $t$. In masked diffusion models, the target distribution is the mask state $\pi = m$. Thus, the forward process can be written as: $q(x_t \mid x) = \text{Cat}(x_t; \alpha_t x_0 + (1 - \alpha_t)\pi)$, where $\alpha_t$ is the corruption schedule (e.g., linear, cosine). To approximate the posterior distribution $q(x_{t-1} \mid x_t, x_0)$, the generative model learns a parameterized backward process: $p_\theta(x_s \mid x_t) \triangleq q(x_s \mid x_t, \mu_\theta(x_t, t))$, where $\mu_\theta(x_t, t) \in \Delta^{m+1}$ is a probability vector parametrized by a neural network $f_\theta$ with a softmax applied to the output logits (note the $m$-th output is forced to 0 since the clean data cannot be masks): $\mu_\theta(x_t, t) = \text{softmax}(f_\theta(x_t, t)) \cdot \mathbf{1}_{\{x_t = m\}} + x_t \cdot \mathbf{1}_{\{x_t \neq m\}}$. This backward process enables iterative denoising from the fully masked state back to the original sequence $x_0$.

## 2.2 Accelerating Discrete Diffusion Language Model

Masked diffusion models decode the entire sequence in parallel. While this improves decoding parallelism, the per-step computation cost is large, making inference speed difficult to match autoregressive models. To alleviate this issue, Block Diffusion (Arriola et al., 2025) combines fully parallel diffusion decoding with autoregressive decoding. The core idea is to perform autoregressive decoding across blocks while applying diffusion decoding within each block. Several variants have been proposed: Set Block Diffusion (Gat et al., 2025) fine-tunes pretrained autoregressive models into block-wise decoders. ESO-LLM (Sahoo et al., 2025) extends block diffusion by supporting mixed decoding strategies within blocks. CtrlDiff (Huang & Tang, 2025) leverages reinforcement learning to dynamically adjust block length. D2F (Wang et al., 2025) enables pipeline parallelism across blocks, where the decoding of the next block can begin before the previous one finishes.

In addition, original masked diffusion models are not compatible with KVCache. Recent work has explored different cache management strategies, mostly under block-wise semi-autoregressive decoding. For example: Fast-dLLM (Wu et al., 2025) adopts a dual-cache design for efficient reuse. Sparse-dLLM (Song et al., 2025a) investigates dynamic cache eviction. dLLM-KVCache (Liu et al., 2025) designs a feature similarity guided prompt caching.

Most prior efforts have concentrated on reducing inference latency. By contrast, our work takes a different perspective: *exploiting the intrinsic parallelism of diffusion models to improve throughput rather than merely lowering latency*.

## 3 Parallel-Scan Based Decoding

A key distinction between diffusion and autoregressive models lies in the ability of diffusion models to decode in parallel. However, existing decoding paradigms face a trade-off: approaches with high parallelism often suffer from prohibitive computational cost, while those with lower cost achieve limited parallelism. In this section, we argue that the root cause lies in the underutilization of **associative states**. A simple probing experiment is designed to show the existence of this new type of states. Leveraging this insight, we propose a **parallel-scan based structured decoding** method that achieves logarithmic computational depth, balanced parallelism, and a rich design space. Furthermore, we introduce two improvements—*local remasking* and *global aggregation*—and provide a systematic exploration of the resulting decoding design space.

### 3.1 Preliminaries

We begin with a review of existing decoding methods. As shown in Figure 1(b), standard diffusion models denoise all tokens according to a time schedule. Each token must interact with the full KVCache, leading to large per-step cost. Block diffusion in Figure 1(c) extends this framework by introducing block-wise autoregressive decoding: diffusion decoding is applied within blocks, while autoregressive dependencies are enforced across blocks. The decoding of each block is conditioned on the preceding block. This scheme provides two benefits: flexible sequence lengths and cache reuse. However, autoregressive dependencies across blocks introduce positional dependency, which prevents fully parallel decoding.

### 3.2 Motivation

Our goal is to enhance the model's parallel decoding capability. Inspired by classical system design, when attempting to break a sequential bottleneck, the first step is to identify associative

states—states in which the decoding order of tokens can be interchanged. To this end, we investigate the associativity of token decoding order. We design a simple probing experiment: given a prompt, we randomly select a token position as the target. We then choose another token that, under normal decoding, would appear earlier than the target. By modifying this token's vocabulary ID and continuing the decoding process, we observe whether the target token can still be correctly generated. As shown in Figure 2, we find that most token pairs exhibit no mutual influence (blue), while only a small fraction interfere with each other (red). This observation provides a concrete intuition for our system design.

### 3.3 ASSOCIATIVE STATES

In the previous experiment, we observed that diffusion models contain many associative states. A natural question is how to leverage these states for system-level acceleration. Consider a simple illustrative example in Figure 3(c): suppose the pair of blocks 2 and 3 are associative with respect to the states of blocks 0 and 1. In this case, a faster decoding strategy becomes possible: Decode blocks 0 and 2 in parallel, yielding a speculative result $x_2'$. Decode block 3 conditioned on $x_2'$, while simultaneously decoding block 1 from block 0. Once block 1 completes, recompute block 2 conditioned on block 1. Under the standard autoregressive regime, this process would require $4B$

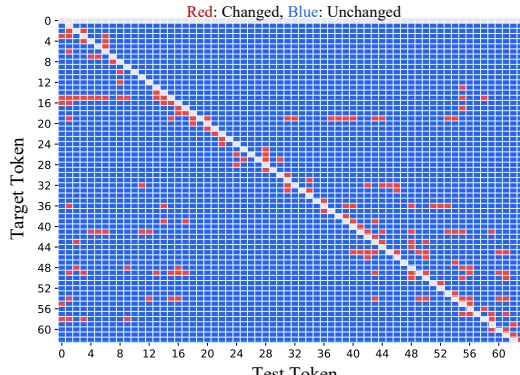

Figure 2: Token Influence Graph. Red points indicate cases where the decoding of the target token is altered, while blue points indicate cases where the target token remains unaffected.

steps, where $B$ is the block size. By exploiting associative states, however, the proposed strategy reduces decoding to $3B$ steps. Step 2 resembles *self-speculative decoding*, while Step 3 performs *self-refinement*. Unlike semi-autoregressive decoding, which irreversibly unmasks tokens, this approach admits dynamic refinement, illustrating the rich and largely unexplored decoding behaviors.

### 3.4 PREFIX NETWORKS AND DECODING BEHAVIORS

A prefix network establishes a structured mapping between network topology and decoding behavior. Each node corresponds to a decoding block, and edges encode the dependency relations among blocks. The structure of the prefix network dictates how information is aggregated across different levels of decoding. We highlight two fundamental types of branches:

**Bypass Branch and Global Aggregation** As illustrated in Figure 3(b), the bypass branch introduces an auxiliary path that allows a block to incorporate global contextual information before local refinement. At the beginning of each decoding level, tokens are decoded with access to the full KV-Cache (global view), ensuring that contextual dependencies are preserved. Subsequently, tokens are refined using the block-local cache. This mechanism prevents information loss caused by premature masking and provides a safeguard for stability across levels.

**Direct Branch and Local Remasking** The direct branch propagates partially decoded results directly across levels without bypassing global aggregation. It enables lightweight forward propagation of intermediate states, allowing faster resolution of tokens while preserving the hierarchical dependency structure. At the end of each level, a global aggregation step compares confidence scores between the direct branch results and bypass branch results. Tokens with higher confidence are retained, while others are remasked for refinement in subsequent levels.

Together, the bypass and direct branches form a complementary mechanism: the bypass branch ensures robustness through global conditioning, while the direct branch enables efficient propagation of intermediate hypotheses. This dual-branch structure provides both stability and efficiency, making prefix networks well-suited for structured parallel decoding.

### 3.5 DESIGN SPACE

Our design space spans three key dimensions: **prefix-network topology**, **cache strategy**, and **parameter configuration**.

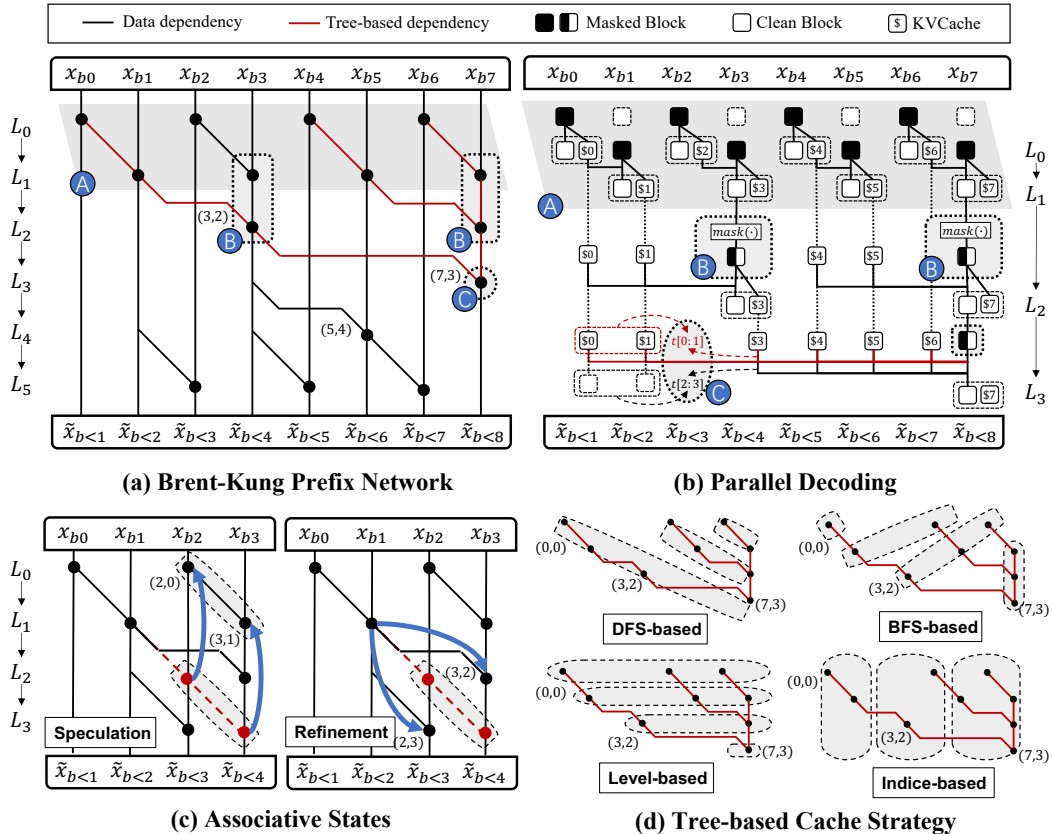

Figure 3: Illustration of parallel-scan decoding. (a) Each vertical line corresponds to a block position $x_{b_i}$, and horizontal levels $L_0, L_1, \ldots$ denote decoding stages, enabling prefix-scan style decoding with logarithmic depth. (b) Decoding process of the prefix-network: A. parallel decoding; B. local remasking; C. global aggregation. (c) Illustration of self-speculation and self-refinement. (c) Four types of cache strategies.

**Prefix-Network Topology.** We explore several classic prefix-network topologies, including Brent–Kung, Kogge–Stone, and Sklansky. These topologies differ in decoding trade-offs such as memory efficiency and work efficiency. Beyond complexity, they also affect the update order of the KVCache. As shown in Figure 4, although all three patterns attempt to reconstruct the autoregressive causal mask, their aggregation priorities differ: Brent–Kung tends to prioritize horizontal recovery of attention scores, Sklansky emphasizes vertical recovery, while Kogge–Stone favors diagonal recovery. Such structural biases imply different memory access and synchronization behaviors.

**Cache Strategy.** We design a tree-based cache framework that dynamically selects key–value (KV) entries depending on the decoding stage. Specifically, we consider four selection modes: **DFS-based**: depth-first traversal, prioritizing deeper dependencies before lateral ones. **BFS-based**: breadth-first traversal, updating cache level by level. **Level-based**: reusing KV entries according to prefix-network layers, enabling synchronous updates across nodes at the same level. **Indice-based**: selecting KV entries based on token indices, which allows fine-grained control and conditional reuse of specific positions.

**Parameter Configuration.** In addition to topology and cache strategy, several parameters critically affect decoding efficiency: **Remask ratio** between consecutive levels, controlling how aggressively uncertain tokens are remasked. **Block size**, which determines the granularity of block-wise diffusion. A summary of the parameter space is provided in Table 1.

**Search Method.** To explore the design space, we adopt GSM8K as a proxy dataset. We sample 600 random configurations, and for each configuration evaluate about 80 samples. The mean latency

Table 1: Search space of prefix-network design. We summarize topology options, cache strategies, and tunable parameters with their corresponding ranges.

| Category | Description | Values |
|---|---|---|
| Prefix-Network Topology | Structural pattern of prefix aggregation, affecting KV-cache update order and synchronization behavior. | {Brent–Kung, Kogge–Stone, Sklansky} |
| Cache Strategy | Mode of selecting and updating KV entries during decoding. | {DFS-based, BFS-based, Level-based, Indice-based} |
| Remask Ratio | Fraction of tokens re-masked between consecutive levels, controlling aggressiveness of correction. | {0.1, 0.5, 0.9} |
| Block Size | Granularity of block-wise diffusion during decoding. | {4, 8, 16} |
| Global Merge Steps | Frequency of global aggregation across levels (as fraction of span). | {0.25, 0.5, 0.75} |
| Confidence Threshold | Token acceptance threshold controlling speculative decoding stability. | {0.5, 0.7, 0.9} |

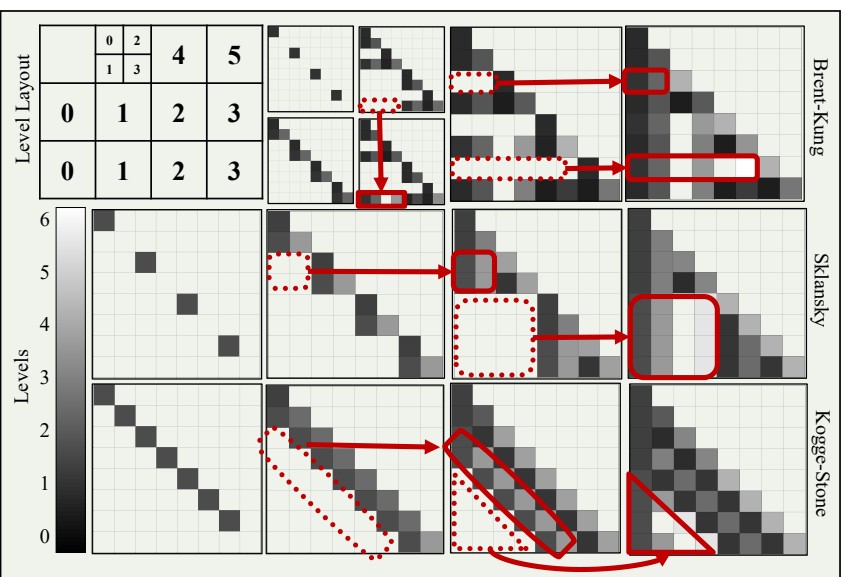

Figure 4: Attention patterns induced by three prefix-network topologies: Brent–Kung (horizontal filling), Sklansky (vertical filling), and Kogge–Stone (diagonal filling). Red arrows indicate how dependencies are progressively reconstructed across levels.

and performance (measured as pass rate) are combined into a unified score. The configuration with the highest score is selected as the final scheme.

## 4 EXPERIMENTS

We empirically validate *parallel-scan based structured decoding* along three axes that mirror the method design: (i) **topology** (Brent–Kung, Sklansky, Kogge–Stone); (ii) **global & local mechanisms** (global aggregation and local remasking); and (iii) **configuration** (confidence threshold, remask ratio, merge step).

### 4.1 EXPERIMENTAL SETUP

**Models and Datasets.** We evaluate on LLaDA-8B-Instruct as the baseline model and consider four tasks that reflect arithmetic, symbolic, and program-synthesis behaviors: GSM8K, MATH,

Table 2: Comparison under two remasking strategies. We report scores on GSM8K, MATH, HumanEval, and MBPP benchmarks for two generation lengths (256/512). LLaDA-8B-instruct (Nie et al., 2025) is the shared baseline; 'Rand' and 'LC' are two variants of the local remasking strategy, representing the random remasking and low-confidence remasking strategies, respectively. Higher is better; **green bold** marks the best within each (length, dataset) group.

| Dataset | Gen_length = 256 | | | Gen_length = 512 | | |
|---|---|---|---|---|---|---|
| | LLaDA | Ours (Rand) | Ours (LC) | LLaDA | Ours (Rand) | Ours (LC) |
| GSM8K (Cobbe et al., 2021) | 58.2 | 61.2 | **62.6** | 27.8 | 56.4 | **60.7** |
| MATH (Hendrycks et al., 2021) | 24.1 | **25.9** | 25.6 | 18.1 | 25.5 | **25.8** |
| HumanEval (Chen et al., 2021) | **39.6** | 25.6 | 29.9 | **43.9** | 26.2 | 29.3 |
| MBPP (Austin et al., 2021) | **36.2** | 27.8 | 27.6 | **34.8** | 31.4 | 31.0 |

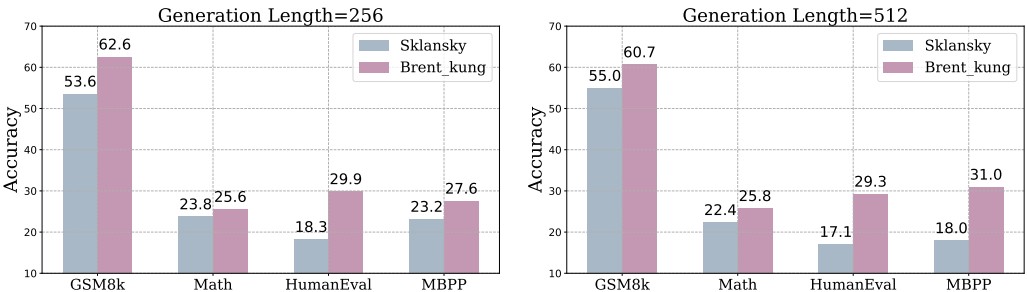

Figure 5: Performance comparison between Sklansky and Bren_kung patterns under different generation budgets.

HumanEval, and MBPP. We report accuracy on GSM8K/MATH and Pass@1 on HumanEval/MBPP. Generation lengths are set to 256 or 512 tokens. Following the official default settings (Nie et al., 2025), we evaluate the 5-shot performance on GSM8K (Cobbe et al., 2021), 4-shot performance on MATH (Hendrycks et al., 2021), 0-shot performance on HumanEval (Chen et al., 2021), and 3-shot performance on MBPP (Austin et al., 2021). We compare the baseline model with our structured parallel decoder under two local remasking variants: **Rand** (random) and **LC** (low-confidence). Throughout, the parallel-scan decoding follows the prefix-network schedule as described in Sec. 3, where global aggregation performs confidence-based selection, while local remasking is responsible for identifying uncertain tokens and remasking them for refinement at the next level. Note that in our experiments, LLaDA is implemented in a full-block diffusion generation mode to align with our globally parallel decoding setting. That is, instead of performing LLaDA inference in a semi-autoregressive decoding mode, we set its step number and block size to the full generation length, allowing both the baseline model and our structured decoder to decode tokens across the sequence at any step.

**Metrics.** Accuracy or Pass@1 is reported for quality evaluation on each benchmark, while the inference efficiency is measured by average throughput (tokens/s). In addition, we further analyze confidence dynamics to probe how associative states are exploited across levels.

**Hyperparameters and Search Space.** We sweep confidence thresholds $\{0.5, 0.7, 0.9\}$, remask ratios $\{0.1, 0.5, 0.9\}$, and global merge steps $\{0.25, 0.5, 0.75\}$ (fraction of the level span between merges). The complete search space is detailed in Table 1.

## 4.2 MAIN RESULTS

Table 2 summarizes the performance under two generation budgets, where there are 4 observations:

**Reasoning benefits most from structured parallelism.** On GSM8K, the LC variant improves the baseline by +4.4 at 256 (58.2→62.6) and by a large margin of **+32.9** at 512 (27.8→60.7). On MATH, gains are consistent yet modest: +1.5 at 256 and +7.7 at 512. These trends align with the probing evidence in Figure 2: most token pairs are *associative* (non-interfering), which our prefix-scan schedule exploits by decoding many blocks in parallel and only revisiting the small fraction of order-sensitive positions through *local remasking* and *global aggregation*. As the length budget grows to 512, the benefit of logarithmic-depth decoding compounds, yielding larger gains.

**Programming tasks show a trade-off with the structured output requirement.** For HumanEval/MBPP, the quality drops (e.g., HumanEval 39.6→29.9 at 256, 43.9→29.3 at 512), which

Table 3: Throughput comparison between baseline model and our methods under different generation budgets.

| Dataset | Gen_length = 256 | | | Gen_length = 512 | | |
|---|---|---|---|---|---|---|
| | LLaDA | Ours | Accelerate | LLaDA | Ours | Accelerate |
| GSM8K | 19.5 | 50.1 | **2.57×** | 16.1 | 68.4 | **4.25×** |
| MATH | 26.5 | 46.8 | **1.83×** | 21.3 | 53.5 | **2.50×** |
| HumanEval | 28.9 | 42.1 | **1.46×** | 28.0 | 57.3 | **2.05×** |
| MBPP | 25.1 | 62.0 | **2.47×** | 20.7 | 61.9 | **3.00×** |

Table 4: Performance comparisons between decoding with and without the self-refinement process.

| Dataset | Gen_length = 256 | | | Gen_length = 512 | | |
|---|---|---|---|---|---|---|
| | W/O SR | W/ SR | Δ Accuracy | W/O SR | W/ SR | Δ Accuracy |
| GSM8K | 50.4 | 53.6 | **+3.2%** | 52.7 | 55.0 | **+2.3%** |
| MATH | 22.6 | 23.8 | **+1.2%** | 21.4 | 22.4 | **+1.0%** |
| HumanEval | 14.6 | 18.3 | **+3.7%** | 15.9 | 17.1 | **+1.2%** |
| MBPP | 23.6 | 23.2 | **-0.4%** | 18.0 | 18.0 | **0.0%** |

is reasonable due to the strict programming rules for the verification. However, Sec. 4.3 shows that *self-refinement* (Figure 3c) recovers a significant portion of the loss, indicating that code exhibits sparser associativity with long-range, order-sensitive constraints. These constraints make parallel speculation helpful for throughput, but they require stronger correction to match the base model's carefully staged autoregressive flow.

**Topology matters: Brent–Kung dominates the quality–throughput frontier.**  Across the space, Brent–Kung consistently yields the best Pareto points (Figure 4). Its horizontal-first reconstruction preserves early block semantics before deep aggregation and matches the empirical associativity pattern from Figure 2—nearby tokens are largely non-interfering and can be aggregated progressively. Sklansky (vertical-first) and Kogge–Stone (diagonal-first) are competitive but less stable at large depths due to aggressive cross-level fan-out amplifies occasional order-sensitive dependencies.

**Local Remasking vs. Random Remasking**  Comparing **LC** to **Rand** in Table 2, low-confidence remasking yields clear benefits on reasoning (e.g., GSM8K 61.2→**62.6** at 256 and 56.4→**60.7** at 512), while the difference is smaller on MATH. On code tasks, LC is still preferable to Rand but remains below the base model, suggesting that error-prone spans are longer and more tightly coupled; they profit from remasking but also demand more frequent or stronger global aggregation.

### 4.3 ABLATION STUDY AND DISCUSSION

**Efficiency Analysis**  Table 3 shows the average throughput performance comparison between the baseline and our method. It can be observed that parallel-scan substantially increases decoding throughput. Specifically, we observe that for the 256-token generation, the average throughput improves from 25.0 to 50.3 tokens/s (**2.0** ×), while for the 512-token generation, the mean rises from 21.5 to 60.3 tokens/s (**2.8** ×).

These gains materialize because prefix-scan reduces the decoding *depth* to $O(\log L)$ while keeping each level cache-friendly (tree-structured KV reuse), consistent with the mechanism in Figure 3.

**Ablations on Associative States and Self-Refinement**  Figure 3(c) motivates using *self-speculation* followed by *self-refinement* to harness associative states. Table 4 contrasts the performance with and without self-refinement, where it can be observed that performance gain is generally achieved across tasks, confirming that parallel speculation plus confidence-gated correction is essential whenever a minority of positions are order-sensitive.

**Hyperparameter Analysis**  Figure 6 analyzes the sensitivity of decoding performance to each hyper-parameter. It can be observed that: 1. higher thresholds postpone commitment, letting more global evidence flow through the prefix tree before retaining tokens; 2. aggressive remasking is beneficial when early parallel speculation is overly optimistic; 3. More frequent global aggregation (0.25) slightly outperforms, indicating that periodic full-context reconciliation stabilizes the few order-sensitive positions.

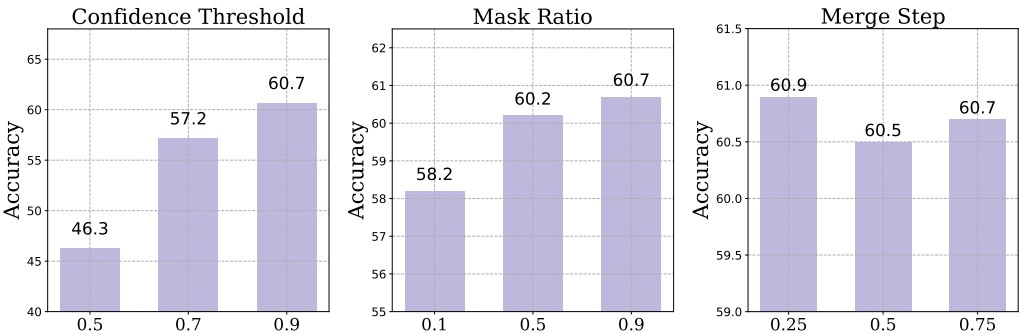

Figure 6: Sensitivity analysis of the three hyperparameters.

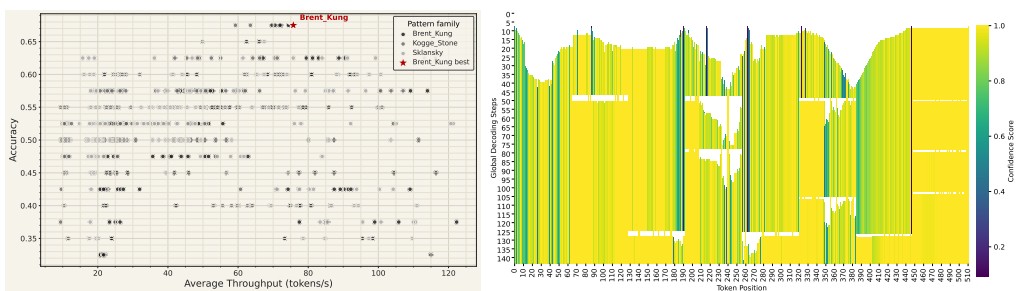

Figure 7: **(left) Design-space search on GSM8K.** Scatter of accuracy vs. throughput for 2400 sampled configurations. Colors denote topology families; the red star marks the best overall setting. Brent–Kung dominates the Pareto frontier. **(right)Confidence dynamics across global steps.** Bright colors indicate high confidence; white spans are (re-)masked by LC. Many tokens stabilize early, while a small set of order-sensitive spans are iteratively corrected via aggregation.

**Design Space Exploration** We perform a random search over 2400 configurations using GSM8K as a proxy; each configuration is evaluated with 25 samples. Figure 7 plots accuracy vs. throughput, colored by topology family. The *Pareto frontier* is dominated by Brent–Kung, with the overall best (red star) lying in the Brent–Kung family and employing LC remasking with a moderate remask ratio and frequent global merges. Kogge–Stone reaches high throughput but is more sensitive to merge cadence (instability at deeper levels), while Sklansky attains mid-range points with fewer outliers. These outcomes echo the qualitative attention reconstructions in Figure 4: horizontal-first recovery (Brent–Kung) respects local associativity and defers reconciliation to controlled aggregation steps.

**Confidence Dynamics and Error-Correction Behavior** Figure 7 visualizes confidence over token positions across global decoding steps. Bright regions (high confidence) expand quickly along the prefix tree; white gaps denote spans re-masked by LC. Two patterns emerge: (i) many early tokens stabilize rapidly and persist across levels (consistent with abundant associativity), and (ii) a small fraction of later spans toggle between masked/unmasked but are progressively resolved after global merges. These dynamics explain the strong gains on reasoning tasks and clarify why code, with denser long-range constraints, benefits from refinement.

## 5 CONCLUSION

We presented parallel-scan-based structured decoding, a simple yet effective framework that exploits associative states to reduce decoding depth to $O(\log L)$ while retaining cache efficiency. Two mechanisms—local remasking and global aggregation—turn speculative parallelism into reliable predictions, and a systematic exploration of prefix-network topologies reveals robust design choices. Key notes are: (1) Parallel-scan delivers **2.0–2.8×** average throughput gains (up to **4.2×**) with preserved or improved accuracy on reasoning tasks. (2) Confidence-guided remasking coupled with periodic global aggregation consistently stabilizes quality by focusing correction on the few order-sensitive positions. (3) Among classic prefix networks, **Brent–Kung** dominates the quality–throughput frontier, matching the empirical associativity pattern observed in our probing study.

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

# 6 APPENDIX

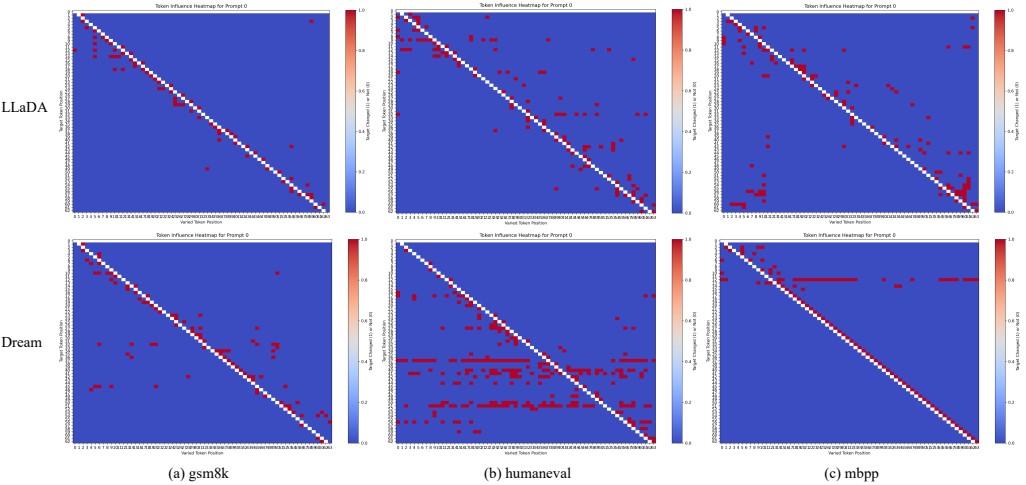

(a) gsm8k     (b) humaneval     (c) mbpp

Figure 8: Token Influence Graph. Red points indicate cases where the decoding of the target token is altered, while blue points indicate cases where the target token remains unaffected. X axis represents the position of test token (up: 0, down: 64) and Y axis represents the position of target token (left: 0, right: 64).

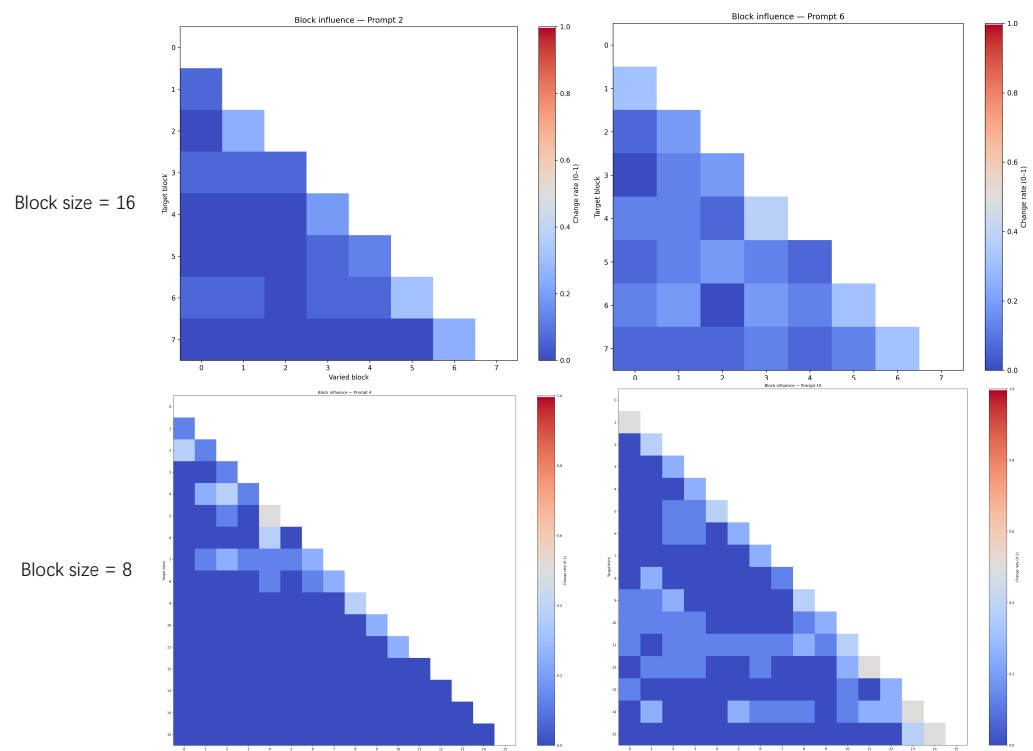

Figure 9: Block Influence Graph. Red points indicate cases where the decoding of the target block is altered, while blue points indicate cases where the target block remains unaffected. The base model is LLaDA. The dataset is HumanEval.

