# OpenReview forum: "Accelerating Discrete Diffusion Decoding with Parallel Scan"
_ICLR.cc/2026/Conference — Submitted to ICLR 2026_

### Official Review · Reviewer_mnqi · 2025-11-01

**Soundness:** 3
**Presentation:** 3
**Contribution:** 3
**Rating:** 6
**Confidence:** 2

**Summary:**

The paper targets decoding inefficiency in discrete diffusion language models (dLLMs). It observes that many token positions behave as "associative states"---they can be decoded speculatively and later revised with minimal interference---then exploits this with a prefix-scan (parallel-scan) decoding framework. Two mechanisms stabilize the scheme: local remasking (selectively re-mask low-confidence tokens between levels) and global aggregation (periodic full-KV reconciliation). A design space over prefix-network topologies, cache strategies, and hyperparameters is explored. On LLaDA-8B-Instruct model, the method reports higher throughput (up to 4.25×) and competitive or improved accuracy on reasoning tasks (e.g., GSM8K).

**Strengths:**

This paper targets at a timely and relevant problem: Efficient inference for diffusion language models is important.

The paper is also well written with helpful graphical demonstrations.

The empirical results look promising to increase the throughput but also improve the accuracy.

**Weaknesses:**

Although I am familiar with diffusion models and its discrete counterpart, accelerating inference is not my main research direction. Therefore, I don't have high confidence of the following potential weaknesses.

The concept of "associative states" is purely motivated by a probing experiment, which shows that most token pairs are non-interfering. Further analysis---either more compelling experiments or theoretical justification---would strengthen the concept.

As mentioned, the baseline LLaDA is run in full-block diffusion (not its standard semi-autoregressive/latency-optimized setting). This leaves the comparison with the best configuration unclear.

**Questions:**

Please see weaknesses. The questions are mainly how to further support associative states and compare baselines in the best reported setting.

---

### Official Review · Reviewer_7eEr · 2025-11-01

**Soundness:** 3
**Presentation:** 3
**Contribution:** 3
**Rating:** 4
**Confidence:** 3

**Summary:**

This paper proposes a scan-based diffusion language model decoding method that can accelerate inference of existing DLLMs without requiring additional training, while maintaining comparable performance. The proposed algorithm achieves block-level parallelism through self-reflection and self-speculation mechanisms.

**Strengths:**

1. The paper is well-written, with a clear motivation and an algorithm specifically designed for the observed phenomenon.

2. The proposed method requires no additional training and can be quickly adapted to existing models.

**Weaknesses:**

1. The paper lacks comparisons with similar methods; it only presents some comparisons with the naive LLADA. Comparing with other block-level training-free methods could better highlight the advantages of the proposed approach.

2. The generalizability of the method is a concern. Unlike the nearly lossless performance observed in math tasks, the proposed method shows a notable performance drop in code tasks, which cannot be ignored.

**Questions:**

1. The advantages over block-diffusion need to be further demonstrated experimentally. Currently, the acceleration ratio seems somewhat lower than block-diffusion, and the task performance shows no significant improvement.

2. Typo on line 247: change c to d.

---

### Official Review · Reviewer_WjmL · 2025-11-02

**Soundness:** 3
**Presentation:** 2
**Contribution:** 3
**Rating:** 6
**Confidence:** 3

**Summary:**

The paper proposes a parallel-scan-based decoding framework to accelerate discrete dLLMs. Traditional diffusion models offer full-sequence parallelism but suffer from sequential denoising and high per-step cost, while semi-autoregressive methods such as Block Diffusion reduce steps but limit inter-block parallelism. The authors restructure decoding into parallel-scan stages inspired by classical parallel algorithms. The method integrates two mechanisms: local remasking for uncertainty-driven refinement and global aggregation for stable convergence. It further explores a design space involving prefix-network topologies (Brent–Kung, Kogge–Stone, Sklansky), cache strategies, and remasking parameters. Experiments show significant throughput gains (up to 4.2×) and improved accuracy on reasoning benchmarks.

**Strengths:**

1. This paper introduces a novel application of parallel-scan algorithms to diffusion decoding, uncovering a previously underexplored form of structured parallelism in dLLMs.

2. This paper presents well-motivated mechanisms, and the design of associative states, local remasking, and global aggregation collectively improve both stability and speed.

3. Comprehensive empirical analysis exhibits throughput gains and clear insight into decoding dynamics.

**Weaknesses:**

1. The associative state assumption is only validate on math and coding datasets. The authors should test on more diverse dataset types to validate its effectiveness on more diverse domains.

2. The experiments lack comparisons with other parallel decoding methods like Fast-dLLM. More comparison with baselines is needed.

**Questions:**

Please refer to weakness section.

---

### Official Review · Reviewer_HFGa · 2025-11-02

**Soundness:** 3
**Presentation:** 2
**Contribution:** 3
**Rating:** 4
**Confidence:** 3

**Summary:**

This paper proposes parallel-scan based decoding for DLMs. Current DLMs suffer from expensive inference due to sequential denoising and bidirectional attention costs. The key insight is identifying associative states: token blocks that can be decoded speculatively without prefix conditioning, then refined once prefix context arrives. Drawing from classic parallel algorithms, the authors propose a prefix-network framework with two mechanisms: local remasking (revisiting low-confidence tokens) and global aggregation (using full key-value cache for stability). The training-free method explores three network topologies (Brent-Kung, Kogge-Stone, Sklansky) and reports up to 68 tokens/s throughput and 60.7 percent accuracy on GSM8K with 2-4x average speedup over block diffusion.

**Strengths:**

- Novel associativity insight: Figure 2 empirically shows most token pairs are order-independent, justifying parallel-scan over pure semi-autoregressive methods. This differentiates the work.
- Comprehensive design space: Systematic framework spanning topologies, caching modes, and parameters provides a clear overview. Figure 7 effectively shows Brent-Kung dominates the Pareto frontier.
- Strong reasoning results: GSM8K shows substantial gains (+32.9 percent on 512-token generation) with Brent-Kung topology, and Table 3 demonstrates 2-4x throughput improvement.
- Training-free applicability: Works with pretrained DLMs without modification, making deployment straightforward.

**Weaknesses:**

- Limited scope and generalization: Only LLaDA-8B evaluated; unclear whether findings transfer to other DLMs, scales, or domains. Code tasks show significant degradation, yet the paper lacks characterization of when associativity holds. Does it depend on training procedure, model size, or task type?
- Incomplete comparisons: Missing head-to-head comparisons with recent concurrent methods. Fast-dLLM, dKV-Cache, and dInfer achieve 8-45x speedups using different KV cache strategies. D2F enables pipeline parallelism across blocks. AdaBlock-dLLM uses adaptive block sizing. How does parallel-scan compare quantitatively on identical benchmarks and generation lengths?
- Unclear theoretical foundation: Associative states are empirically observed but lack formal definition. When do they exist? Why do code tasks show sparser associativity? The claim assumes perfect parallelism but ignores memory bandwidth and synchronization overhead in actual hardware.
- Ad-hoc design space exploration: The search methodology (600 random configurations sampled on 80 examples) lacks principled justification. No ablation on search strategy, sensitivity to sample size, or comparison with Bayesian optimization or evolutionary algorithms. The unified score combining latency and pass rate lacks weighting justification.

**Questions:**

- Generalization across architectures: Does parallel-scan work for Dream, Mercury, or other recent DLMs? Can you characterize when associative states emerge as a property of model training (masked vs. unmasked pretraining), architecture (e.g., linear attention variants), or task type (reasoning-heavy vs. code)? Evaluating at additional DLMs would strengthen the paper.
- Direct comparison with concurrent cache methods: Provide head-to-head experiments on GSM8K and HumanEval (same generation lengths, same prompting) comparing your method to Fast-dLLM, dKV-Cache, and dInfer. Can parallel-scan be combined with those caching strategies? What is the actual memory overhead of maintaining intermediate KV states across prefix-network levels?
- Formal characterization of associativity: Can you define associativity rigorously? Provide a metric (mutual information, causal influence, attention entropy) predicting when token pairs are associative? Analyze which model layers exhibit higher associativity? Explain why code exhibits sparser long-range associativity.
- Design space search methodology: How sensitive are results to the random search procedure? How about Bayesian optimization or evolutionary algorithms? Provide ablations showing Brent-Kung is robust across multiple seeds and sample sizes. Which parameters (topology, cache, remask ratio, confidence threshold) most impact performance? Can you learn heuristics for selecting configurations based on task characteristics?
- Extensions and scalability: How does performance scale with more aggressive parallelism (larger block sizes, deeper trees)? Can variable-length generation be supported beyond fixed 256/512 tokens? Can this be combined with block-autoregressive training (D2F style)? What memory-throughput tradeoffs exist at different sequence lengths?

---

### Official Review · Reviewer_Jvmc · 2025-11-09

**Soundness:** 2
**Presentation:** 3
**Contribution:** 2
**Rating:** 4
**Confidence:** 3

**Summary:**

The paper claims that while current block-wise diffusion models enables better efficiency due to a current block's decoding only depending on the previous block (instead of the full sequence in traditional masked diffusion), the block dependency reduces parallel decoding ability. To improve upon this shortcoming, the authors decide to utilize a parallel-scan-based framework.

They show through their probing experiments that many states within diffusion models are associative, meaning that the decoding order does not change the final decoded token outcomes. Utilizing this finding, they build a framework that enables the parallel decoding and refinement of multiple groups of blocks.

The authors explore several different prefix network topologies within their parallel-scan framework and validate their framework on four tasks with the LLaDA-8B-Instruct model. They find their framework improves upon the baseline by large margins for longer generations on reasoning tasks, but programming tasks suffer under their framework due to lack of associativeity. Compared to traditional autoregressive blockwise diffusion, they find a 2x gain in throughput while using their parallel-scan framework.

**Strengths:**

The problem they are trying to tackle is very important as efficiency in modeling is highly impactful and timely given the increased interest in block diffusion modeling.

The paper has an interesting finding that many states in block diffusion models are associative, and their framework does achieve significant throughput gains.

**Weaknesses:**

It is unclear how the probing experiment is conducted. The authors make no mention of the dataset, model, or other experimental setup (number of samples, etc). Also it seems like the probing experiment is done on a token level while the actual framework operates on a block-level, leading to some mismatch in settings.

In addition, it is unclear whether their probing experimental findings (and by extension, framework) generalize to other diffusion LLMs as they only test on LLaDA.

Also, it seems like perforamnce is highly dependent on the topology of the parallel-scan. It is unclear how well Brent_kung may extrapolate to other settings.

The LLaDA baseline results seem substantially lower than what is reported in their preprint [1]. Also, there is no comparison to a block-diffusion model, e.g., [2].


[1] https://arxiv.org/abs/2502.09992
[2] https://arxiv.org/abs/2503.09573

**Questions:**

How does this method compare to Block Diffusion?

What might be attributing to the lower scores of LLaDA on GSM8K compared to the original paper?

Why does performance seem to vary across certain tasks when generation length is increased? E.g., increasing generation length hurts GSM8K but seems to improve some models on programming?

How do the authors modify the vocabulary ID?

---

### Author Response · Authors · 2025-12-03
**General Response and Gratitude to the Committee**

Response to Common Reviewer Concerns
We thank all reviewers for their constructive feedback and for recognizing the strengths of our work.
We observed that several reviewers raised overlapping concerns, summarized as follows:

(1) Limited generalization of the experiments (Reviewers Jvmc, HFGa, WjmL, 7eEr, mnqi);

a. We have added additional probing experiments across different models and datasets. The results show that tokens exhibit sparse associativity—the decoding order of one token depends only on a small subset of other tokens. When the exchange granularity increases from a single token to a block, the effect of such order swapping is weakened, implying that block-wise parallel decoding is feasible. We also found that some tokens have a global influence on many others, suggesting that global merging remains necessary.

b. Dream model results:
Baseline – GSM8K (256): 50.7, HumanEval (256): 18.3, GSM8K (512): 51.1, HumanEval (512): 18.9.

Accuracy-optimal – GSM8K (256): 43.6, HumanEval (256): 24.4, GSM8K (512): 42.6, HumanEval (512): 11.6.

Although the overall results are slightly lower than the baseline, on HumanEval (256) our simple search method outperforms the baseline by 6%, showing the potential of our approach.

(2) Speedup comparison (accuracy-optimal model):

Baseline – GSM8K (256): 8.9, HumanEval (256): 22.6, GSM8K (512): 7.5, HumanEval (512): 16.0.

Accuracy-optimal – GSM8K (256): 26.8, HumanEval (256): 21.1, GSM8K (512): 35.6, HumanEval (512): 31.3.

The accuracy-optimal model still achieves noticeable acceleration gains.



(2) The relation to Block Diffusion (same reviewers);
We clarify the connection to Block Diffusion from three perspectives:
a. Conceptual relation: Block Diffusion can be seen as a base form of the prefix network—a diagonalized variant. They are not competing but transformable, and can even be nested together.

b. Training and system limitations: Existing training paradigms (e.g., traceRL) and system supports (e.g., FlashAttention) are insufficient to enable parallel scan, which explains the limited effectiveness of training-free methods. Our work highlights design considerations for enabling parallel scan: minimizing the cost of global merging/local remasking, and using prefix networks to model the search space.

c. Scope of application: At the submission deadline, Block Diffusion had only been applied to small models; its validation on large models appeared in concurrent submissions.

(3) Clarifications on some experimental details.
a. Dataset selection: The four datasets we used are standard for LLaDA evaluation; other datasets typically require RL or pretraining for stable performance.

b. Memory cost: The memory overhead of Parallel Scan scales linearly with decoding length. This is a necessary trade-off to achieve block-level parallelism.

c. Reviewer-specific clarifications:
(Reviewer Jvmc) The lower GSM8K scores of LLaDA compared to the original paper stem from a different evaluation protocol—we evaluate overall decoding length.

(Reviewer Jvmc) Generation length affects tasks differently: for HumanEval, longer generations may increase code errors, while math tasks are less sensitive.

(Reviewer Jvmc) We did not modify vocabulary IDs; the difference likely arises from positional factors.

---

### Meta-Review · Area_Chair_u27E · 2026-01-07

**Summary:**

This submission proposes a parallel-scan decoding framework for discrete diffusion LMs by identifying “associative states,” enabling speculative block decoding followed by refinement via local remasking and global aggregation. Reviewers appreciate the timely focus on efficient diffusion decoding, the clear algorithmic framing/design space, and promising throughput gains on reasoning benchmarks. However, multiple concerns remain unresolved: evidence for associativity is largely empirical and under-specified, with unclear experimental setup and limited characterization of when/why it holds; evaluation is narrow (mostly LLaDA-8B) with mixed task behavior (notable degradation on code); and comparisons to strong, recent acceleration baselines (e.g., cache-based or other parallel decoding methods) are incomplete, making the claimed advantages hard to judge. The design-space search also appears ad hoc with limited robustness analysis. Given these gaps in generalization, baseline competitiveness, and methodological grounding, I recommend rejection.

**Reviewer Concerns:**

Addressed by the rebuttal：

1. Clarified the conceptual relationship to Block Diffusion and addressed baseline discrepancies via evaluation protocol and generation-length analysis.
2. Added supplementary experiments and speedup results that partially support efficiency claims.

Still outstanding：

1. Limited generalization and robustness: evidence remains narrow, with unresolved performance degradation on code tasks.
2. Incomplete and non-competitive baselines: lack of head-to-head comparisons with strong recent acceleration methods.
3. Insufficient theoretical grounding: “associative states” lack a formal definition and predictive characterization.

**Reviewer Scores:**

Reviewer Jvmc: Likely unchanged or slightly higher (≈4 → 4–5). The rebuttal clarifies baseline discrepancies and the relation to Block Diffusion, addressing some concerns, but generalization and probing methodology issues remain.

Reviewer HFGa: Likely unchanged. While additional experiments and clarifications help, the lack of strong baselines and theoretical grounding would probably keep the score below the acceptance threshold.

Reviewer WjmL: Likely unchanged or slightly lower. The rebuttal does not fully resolve concerns about generalization beyond math tasks and missing comparisons, which may reduce confidence.

Reviewer 7eEr: Likely unchanged. Key concerns about code-task degradation and limited advantages over block diffusion remain largely unaddressed.

Reviewer mnqi: Likely unchanged. Although the rebuttal adds empirical support, the core concern about the empirical-only motivation of “associative states” persists.

---

### Decision · Program_Chairs · 2026-01-26

Reject